# Non-Destructive Quality-Detection Techniques for Cereal Grains: A Systematic Review

**Yiming Liu** [1,2,3,4,†], **Jingchao Zhang** [5,†], **Huali Yuan** [1,2,3,4], **Minghan Song** [1,2,3,4], **Yan Zhu** [1,2,3,4], **Weixing Cao** [1,2,3,4], **Xiaoping Jiang** [1,2,3,4] **and Jun Ni** [1,2,3,4,*]

1 College of Agriculture, Nanjing Agricultural University, Nanjing 210095, China
2 National Engineering and Technology Center for Information Agriculture, Nanjing 210095, China
3 China Engineering Research Center of Smart Agriculture, Ministry of Education, Nanjing 210095, China
4 Collaborative Innovation Center for Modern Crop Production Co-Sponsored by Province and Ministry, Nanjing 210095, China
5 Nanjing Institute of Agricultural Mechanization, Ministry of Agriculture and Rural Affairs, Nanjing 210014, China
* Correspondence: nijun@njau.edu.cn; Tel.: +86-25-84396593
† These authors contributed equally to this work.

**Abstract:** Grain quality involves the appearance, nutritional, and safety attributes of grains. With the improvement of people's living standards, problems pertaining to the quality of grains have received greater attention. Modern quality detection techniques feature unique advantages including rapidness, non-destructiveness, accuracy, and efficiency in detecting grain quality. This review summarizes research progress of these techniques in detection of quality indices of grains. Particularly, the review focuses on detection techniques based on physical properties including acoustic, optical, thermal, electrical, and mechanical properties, and those simulating sensory analysis such as electronic noses, electronic tongues, and electronic eyes. According to the current technological development and application, the challenges and prospects of these techniques are demonstrated.

**Keywords:** appearance attributes; nutritional attributes; safety attributes; non-destructive detection; physical properties; sensory properties; cereal grains

## 1. Introduction

Grains, containing nutritional ingredients including carbohydrates, proteins, fats, vitamins, and minerals, are daily necessities of human life. However, impurities, unsound kernels, fungal toxins, pesticides, and heavy metal residues pose a risk to human health. With regular economic growth and social progress, consumers have begun to pay more attention to the quality and safety of grains and have increasingly larger demands for high-quality and highly safe grains. The quality attributes of grains acceptable to consumers mainly include appearance attributes (size, shape, and color), nutritional attributes (protein, starch, fat, and vitamin), and safety attributes (contaminants such as mildew, pesticide residues, and heavy metal residues). Detailed and specific requirements for quality indices of grains have been set in Chinese National Standards. For appearance attributes, appearance indices including the color and shape, impurity, and unsound kernels of wheat, maize, paddy, and soybean have been elucidated in detail in the Assistant Atlas of Grain Sensory Inspection [1–4]. As to nutritional attributes, the determination methods for nutrients including proteins [5], starches [6], fats [7], ashes [8], amino acids [9], dietary fibers [10], and trace elements [11] have been specified in the Chinese National Standards for Food Safety. In terms of safety attributes, the maximum residue limits for pesticides [12], maximum residue limits for fungal toxins [13], and maximum levels of contaminants [14] in grains have been listed in the Chinese National Standards for Food Safety. There are also detailed requirements pertaining to the appearance [15–18], nutritional attributes [19–25], and safety attributes [26–29] of grains in international standards.

Grain quality is an important index in the grain circulation process involving the production, storage, trading, and processing. Grain quality detection has always been one of the greatest challenges pertaining to the treatment, processing, classification, and safety guarantees needed in the food industry. Traditionally, grain quality detection is realized through sensory and chemical analyses. However, sensory analysis is time-consuming, inefficient, highly subjective, and susceptible to external interference (influences of physical conditions such as fatigue); chemical analysis is expensive, time-consuming, laborious, and destructive, and requires a laboratory. In recent years, to meet the requirements of modern quality inspection, detection techniques based on physical properties such as acoustic, optical, thermal, electrical, and mechanical properties and sensory features including visual, gustatory, and olfactory features have been developed apace. A bibliometric analysis was used in this paper. References on the state of the art in non-destructive quality-detection techniques for cereal grains were searched through the Web of Science search engine as well as the National Standards Full Text Open System and the websites of International Organization for Standardization to obtain relevant quality standards for cereal grains. The standards are to look up by keywords such as cereal, grain, wheat, rice, pulses, maize, toxin, pesticide, and residue on the system and website. The references of the state of the art were obtained within the last five years in the core collection database of the Web of Science search engine. We searched the references by combining the keywords cereal, grain, and quality with the technical words of physical properties such as near-infrared spectroscopy (NIRS), hyperspectral imaging (HSI), Raman spectroscopy (RS), optical, dielectric, nuclear magnetic resonance (NMR), X-ray, electromagnetic, acoustic, thermal, and mechanical, and sensory features such as electronic eye (E-eye), computer vision, electronic nose (E-nose), electronic tongue (E-tongue), and sensory, respectively. We classified all the references according to the quality as the first condition and the techniques as the second condition, and summarized and analyzed the research purposes, research contents, research methods and research results of each reference. The previous references have focused on the applications of a particular detection method to different objects, or some detection methods for selected objects. In this study, we analyzed and summarized the advantages and disadvantages of the methods of physical properties and sensory features in terms of appearance, nutritional and safety attributes of cereal grains, and explored the techniques that can detect various quality indicators in different application scenarios. Based on the latest technical references we have collected, we have mapped the latest research progress of research institutes [30–52] (Figure 1) and commercial instruments [53–86] (Figure 2).

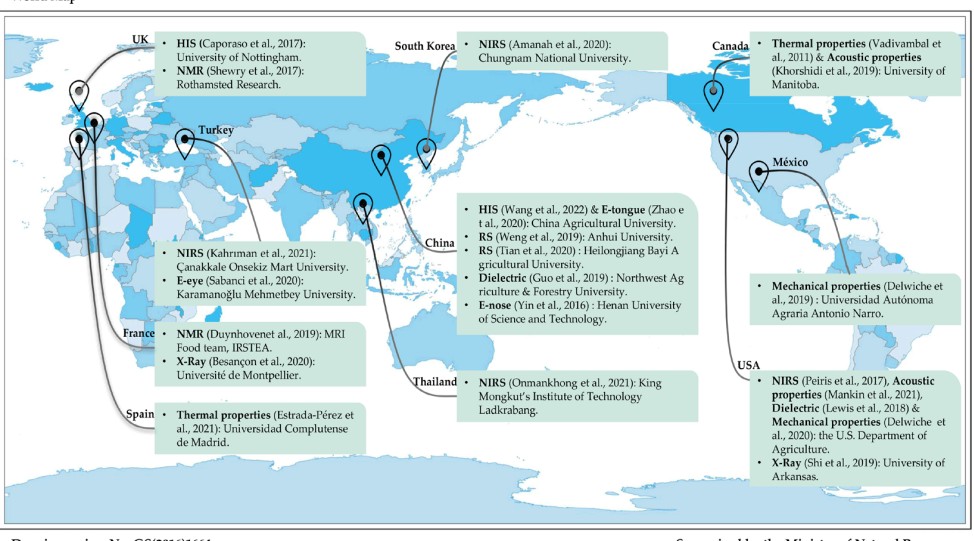

**Figure 1.** The research progress of research institutions on grain quality detection techniques [30–52].

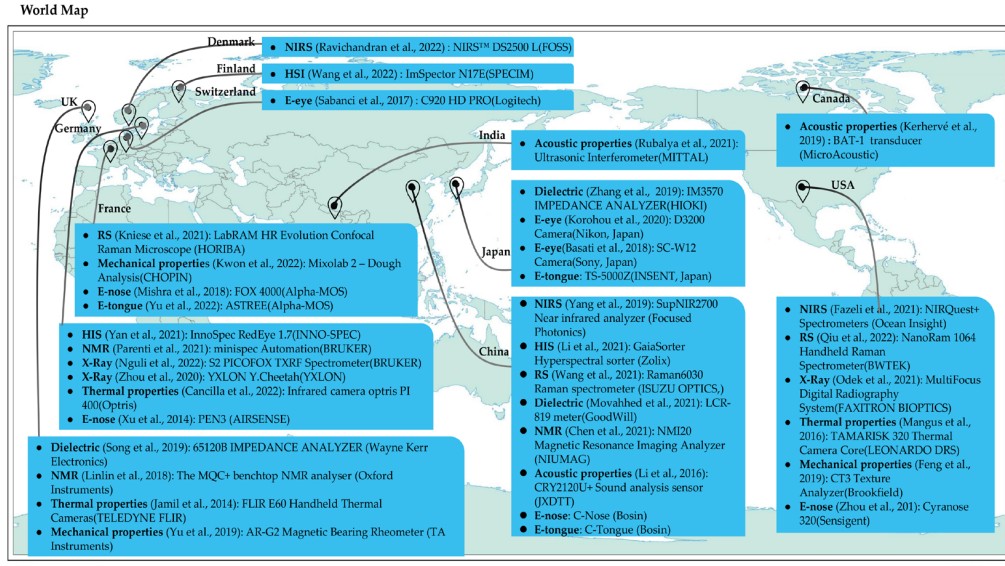

**Figure 2.** Commercial instruments for grain quality detection [53–86].

Among detection techniques for grain quality based on physical properties, those based on optical features such as NIRS, HSI, and RS detect nutritional attributes (proteins, starches, and fats) and safety attributes (pesticide residues and fungal toxins) of grains [87]. The detection is based on the response to light in grains, including its absorption, reflection, transmission, and scattering. In addition, HSI can also be used to determine color and shape [88]; RS also can be adopted to assay heavy metal residues [89]. Detection techniques based on electromagnetic properties such as those based on dielectric properties, NMR, and X-ray realize detection according to response signals of grains in the electrical or magnetic fields. They can be utilized to determine the moisture content [90] and nutritional attributes such as starches and fats [91]. X-rays also can be used to determine trace elements and heavy metals in grains [92]. Detection techniques based on acoustic features were adopted to study nutrients including proteins and appearance attributes such as unsound kernels according to responses of grains to acoustic signals (reflection and transmission) [93]. Detection techniques based on thermal features were used to assess safety attributes involving fungi and appearance attributes including unsound kernels based on differences in thermal radiation of each part of grains [94]. Those based on mechanical properties were employed to study nutritional attributes including proteins and starches according to mechanical features of grains under all kinds of applied load [95]. Among detection techniques for grain quality based on sensory features, E-eyes were used to identify appearance attributes of grains including the color and shape, impurities, and unsound kernels according to features such as color and shape in images [96]. E-noses were employed to evaluate safety attributes involving pesticide residues and fungal toxins and appearance attributes including unsound kernels in accordance with gaseous response signals of volatile organic compounds in grains [97]. E-tongues are often used to detect nutritional attributes such as proteins and starches and safety attributes about heavy metals based on taste response signals in grain leachates [98]. The principles and objects of detection techniques for grain quality based on physical properties and sensory features are listed in Table 1.

**Table 1.** Modern inspection techniques for grain quality.

| Detection methods | | | Principles | Objects | Limitation |
|---|---|---|---|---|---|
| Physical properties | Optical properties | NIRS | Realizing quantitative quality detection and qualitative analysis according to differences in the absorption band and intensity of hydric groups in organic components of grains in the near-infrared region | Nutritional attributes including proteins, starches, and fats, and safety attributes pertaining to pesticide residues and fungal toxins | High precision instruments are expensive and the NIR spectra of different components overlap. |
| | | HSI | Realizing accurate detection of grain quality based on hyperspectral and image data | Nutritional attributes including proteins, safety attributes pertaining to pesticide residues and fungal toxins, and appearance attributes including color and shape | HSI is costly, the amount of hyperspectral data is extremely large, and it is difficult to store and analyze. |
| | | RS | Based on scattering spectra of different components in grains at different light frequencies; achieving quality detection by analyzing molecular vibration and rotation of these components in grains | Nutritional attributes including proteins, and safety attributes pertaining to fungal toxins, pesticide residues, and heavy metals | Fluorescence phenomena on Fourier variation Raman spectral interference, optical systems affecting different vibrational peak overlaps, and Raman scattering intensity. |
| | Electromagnetic properties | Dielectric | According to the response characteristics of grains in the applied electric field | Moisture content | High correlation mainly with moisture. |
| | | NMR | Atomic nuclei with fixed magnetic moments in grains produce a string of response signals with attenuated intensity in the specific impulse trains. | Moisture content and nutritional attributes including starches and fats | Mainly used for moisture state, migration process analysis, high price, complex signal analysis, and imperfect NMR spectrum database. |
| | | X-ray | Elements in grains release X-ray fluorescence of specific energy under X-ray irradiation | Nutritional attributes including trace elements, and safety attributes pertaining to heavy metals | X-ray control is complicated and dangerous. |
| | Acoustic properties | | According to the reflection, scattering, projection, and absorption characteristics of acoustic waves in grains | Nutritional attributes including proteins, and appearance attributes such as unsound kernels | High environmental noise interference. |
| | Thermal properties | | According to differences in thermal radiation of various parts of grains | Appearance attributes such as fungal infection and unsound kernels | High ambient temperature disturbance. |
| | Mechanical properties | | According to the mechanical features of grains under all types of mechanical load | Nutritional attributes including proteins and starches | The association between mechanical properties and quality is unclear. |
| Sensory features | E-eye | | According to features including color and shape in images | Multiple appearance attributes | High requirements for clarity of acquired images, difficulty to identify early mold and pest images, and difficulty to segment multi-seed images. |
| | E-nose | | According to gaseous response signals of volatile organic compounds in grains | Safety attributes pertaining to fungal toxins and pesticide residues | Early mold or mild pesticide residues produce low gas concentrations that are difficult to detect and environmental gas interference. |
| | E-tongue | | According to taste response signals of grain leachates | Nutritional attributes including proteins and starches, and safety attributes pertaining to heavy metals | The detection object is the leachate of seed samples |

The organizational structure of this paper is shown in Figure 3. The article consists of four parts as follows:

(1) Introduction. The introduction of the classification and definition of grain quality (which can be classified as appearance, nutritional, and safety attributes), and the analysis of grain quality detection techniques based on physical properties and sensory features.

(2) Research progress. The current status of the study is divided into three subsections: appearance, nutritional and safety attributes. ① Research progress on detection of appearance attributes of grains based on HSI, X-ray, NMR, thermal properties, acoustic properties, and E-eye. ② Research progress on detection of nutritional attributes of grains based on NIRS, HSI, RS, NMR, X-ray, acoustic properties, mechanical properties, and the E-tongue. ③ Research progress on detection of safety attributes of grains based on NIRS, HSI, RS, X-ray, and the E-nose.

(3) Unsolved technical problems. The problems of high cost, environmental interference, detection principle, limit of detection (LOD), moisture detection, grading and classification of grains and practical application issues in detection applications.

(4) Future research direction. The summary about future research direction, and the methods and attempts to solve the above problems found in some references.

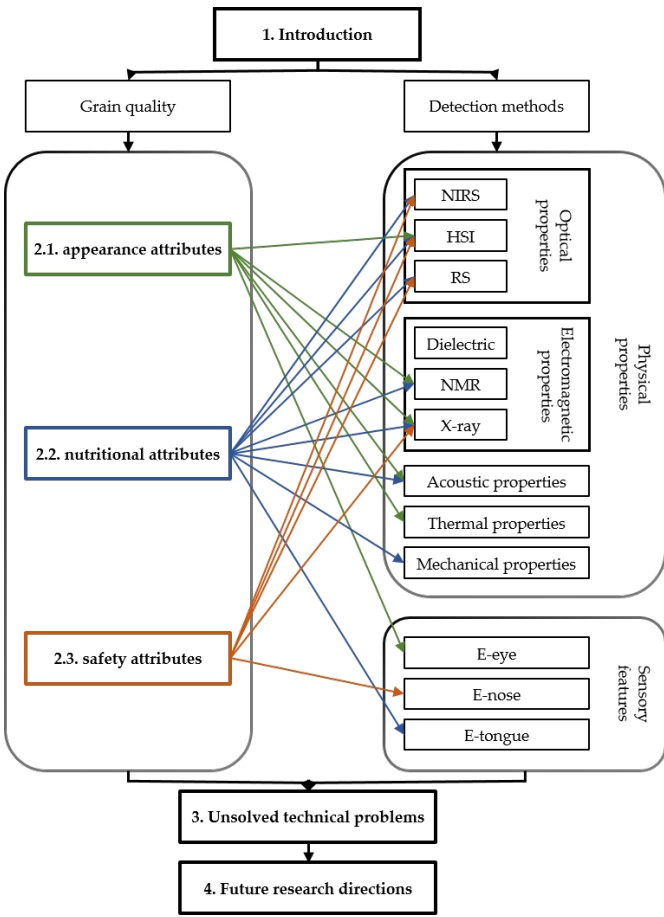

**Figure 3.** Block diagram of the article organization.

## 2. Overseas and Domestic Research Status

*2.1. Non-Destructive Quality Detection Methods for Appearance Attributes of Cereal Grains*

According to descriptions in the Assistant Atlas of Grain Sensory Inspection, the appearance attributes of grains mainly include quality indices such as color and shape, impurities (screen underflows, inorganic impurities, and organic impurities), and unsound

kernels (immature, injured, specked, broken, germinated, and moldy grains). Traditionally, appearance attributes are mainly inspected through an artificial sensory analysis, which is both time- and labor-consuming and greatly affected by subjective factors. Among modern detection methods of appearance attributes, the HSI, NMR, X-ray, those based on acoustic and thermal features, and E-eye all can detect indices of appearance attributes (Table 2).

**Table 2.** Non-destructive quality detection methods for appearance attributes of cereal grains.

| Detection Methods | Objects | Devices | References |
|---|---|---|---|
| HSI | Color and shape, unsound kernels, and impurities | Zolix "GaiaSorter" hyperspectral imaging system | [99–101] |
| NMR | Unsound kernels | NMI20 bench top pulsed NMR analyzer | [102] |
| X-ray | Unsound kernels | Skyscan 1272 X-ray micro-CT scanner | [48] |
| Acoustic properties | Unsound kernels | Self-made impulse signal acquisition device | [103] |
| Thermal properties | Unsound kernels | MLG-II temperature sensor | [104] |
| E-eye | Color and shape, unsound kernels, and impurities | CCD camera or smartphone | [81,105,106] |

HSI, which combines spectral and image information, can more accurately detect the quality information of grains. Li et al. [99] collected the hyperspectral images of wheat grains in four states (healthy, germinated, mildewed, and shriveled) across the range of wavelengths from 866.4 to 1701.0 nm. By using the data enhancement method based on the deep convolutional generative adversarial network, the spectral data are generated, and sample classes are expanded to improve the performance of the classifier. The classification accuracy of the CNN model is increased from 17.50% before expansion to 96.67%. Zhang et al. [100] obtained hyperspectral data from single wheat grains in the reverse and ventral sides at 866.4 to 1701.0 nm and used the principal component loading method to extract characteristic wavelengths. In this way, the SVM model is established for discriminating four types of wheat grains (healthy, germinated, mildewed, and shriveled), and the classification accuracies for the four types of wheat grains all exceed 96%. By collecting the spectra of healthy and *Fusarium*-infected wheat grains at 350 to 2500 nm, Zhang et al. [101] found that the classification accuracy of the spectral classification index method is 0.97.

In NMR detection, the NMR signals of grains and foreign matters therein show significant differences. Shao et al. [102] measured the NMR spectra of wheat grains, Sitophilus zeamais, and their mixture. By analyzing NMR spectra through a combination of biexponential analysis and a discrete method, a series of NMR parameters were obtained, which could be used as diagnostic indices for the number and stage of insect pests in wheat grains. The partial least squares (PLS) results based on NMR relaxation parameters (T21, T22, P21, P22, and A21/22) indicate that the measurement and prediction of insect pests exhibit good correlation ($R > 0.960$).

X-rays can penetrate deeply into a grain and therefore can be used to detect internal structural features thereof. Boniecki et al. [107] observed internal structural damage caused by the granary weevils through digital X-ray images of wheat seeds and developed an artificial neural network model with a three-layer perceptron capable of correctly detecting 100% of infected seeds and 98.4% of healthy seeds. Besançon et al. [48] differentiated between two wheat grain samples based on computerized tomography (CT) data with a resolution of 7 um according to endosperm porosity. They plotted the cumulative distribution curves of pore volume and pore size and analyzed different types of pores to describe endosperm structures. Scanning electron microscopy (SEM) and CT images with the resolution of 1 um can be employed to describe the microstructures of endosperm of durum wheat grains and quantitatively investigate proportions of vitreous endosperm and farinaceous endosperm. Jha and Tripathy [108] used a high-resolution X-ray micro-CT scanner to collect the images of paddy and wheat grains, and the pore sizes of paddy and wheat samples in the CT images are in the range of 0.01 to 0.8 mm$^3$. Their analysis

implies that hybrid solar drying causes compaction of paddy surfaces, cell disruption, and agglomeration of wheat endosperm. By using an X-ray micro-CT system to conduct lossless scanning of whole maize grains, Shao et al. [109] reconstructed the 3D model of the internal structure of maize grains via pseudo-color visualization. The results show that plump maize grains have fewer internal fissures than non-plump grains. Shi et al. [35] developed a fissure detection and measurement algorithm for rough rice kernels based on X-ray images. The algorithm segments individual rice kernels in the X-ray images using the gap-filling method and detects fissures in each rice kernel through adaptive thresholding and the use of a slew of filters. The deviation for the percentage of fissured kernels is less than 2%. Through X-ray scanning and reconstruction, Sun and Xiao [110] conducted visualized analysis on the internal structure of rice at micron resolution. They also determined the volume, shape, and location information of rice chalkiness through 3D reconstruction and volume-based quantification. Zhou et al. [111] scanned wheat grains through micron-resolution X-ray CT and discriminated wheat grains that are infected with mold or not by comparing their grey values in different states. The 3D visualized characterization of the internal structure and pores of the wheat grains was realized according to the difference in attenuation of X-rays, and parameters such as the wheat grain volume, pore volume, and porosity were calculated.

Unsound kernels show different reflection, scattering, projection, and absorption features for acoustic waves. Guo et al. [103] proposed an impact sound signal processing technology based on Gaussian modelling and improved extreme learning machine to detect injured wheat grains. The results indicate that the classification accuracies of the method for uninjured wheat grains and those injured by insect pests and sprouting are 92.0%, 96.0%, and 95.0%, respectively.

Grain and foreign matter therein including insect pests exhibit significant differences in the thermal radiation features. Based on thermal features, the appearance attributes of unsound kernels can be detected. Wang et al. [104] plotted temperature changes on the longitudinal and transverse profiles of bulk maize using cloud images. At different time moments, the areas of six sections and volumes of zones at a temperature above 25 °C were calculated. In addition, the relationship of the area and volume with the temperature difference was revealed. The results show that a large high-temperature zone is formed rapidly at the center of the maize under microbial action. The heat exchange distance in the longitudinal and transverse directions is a function of the temperature difference, and the two have a linear relationship during development of the high-temperature zone. The volume of the high-temperature zone is a cubic polynomial function.

E-eyes, as a non-conventional term, are an image analysis technique in computer vision. An E-eye comprises an image sensor and a pattern recognition system. The image sensor collects image data and the pattern recognition system recognizes and analyzes features such as color and shape in images [96,112,113]. Lingwal et al. [105] used a convolutional neural network (CNN) to classify 15,000 images of 15 wheat varieties based on visual traits. A classification model for wheat varieties under adjustment of different hyper-parameters was established, which indicated a training accuracy and test accuracy of 94.88% and 97.53%, respectively. Basati et al. [81] extracted nine color features, ten shape features, and six statistical features relating to healthy and moth-eaten wheat grains with different moisture contents. Therein, 11 features were selected as the input to artificial neural network (ANN), decision tree, and discriminant analysis classifiers. The results show that the classification accuracy of the J.48 algorithm based on the decision tree is the highest (90.20%), followed by the ANN with an 11-19-2 topology and the discriminant analysis classifier, the classification accuracies of which are separately 87.46% and 81.81%. Based on a multi-class support vector machine (SVM) classifier, Vithu et al. [106] developed a recognition model for the documented type of paddy. The model considers five morphological features and nine color features in paddy images, based on which the overall classification accuracy reaches 90.3%. The average classification accuracies of the model for paddy, organic impurities, inorganic impurities, other paddy varieties, and black rice are 85%, 83.2%, 93%,

98%, and 93.6%, respectively. By analyzing features including the color and shape in grain images, E-eyes can realize the qualitative classification of grain varieties and detection of pests and diseases. Grain investigation methods based on E-eyes are easy and convenient to use and have been extensively applied in quality-detection on grain varieties.

HSI can realize high-accuracy detection of appearance indices by combining image and spectral information. NMR achieves high-accurate detection based on characteristic response signals of insect pests. X-ray technology is used to detect and estimate changes in internal structures of grains and analyze the formation of appearance attributes. The unsound kernels can be detected and discriminated based on differences of normal and unsound kernels in acoustic and thermal features. E-eyes can detect all appearance indices including color and shape, impurities, and unsound kernels of grains.

### 2.2. Non-Destructive Quality Detection Methods for Nutritional Attributes of Cereal Grains

The main nutritional indices of grains include proteins, starches, fats, ashes, amino acids, dietary fibers, and trace elements. Traditionally, nutritional attributes are determined using physical and chemical experimental analysis. The Kjeldahl method for nitrogen determination, hydrolysis, Randall extraction, incineration, amino-acid analyzers, enzymatic gravimetric method, and plasmas are generally adopted to determine protein [5], starch [6], fat [7], ash [8], amino acid [9], dietary fiber [10], and trace element [11] contents in grains. Traditional detection methods generally call for destructive sample preparation and are time-consuming and inefficient. In modern inspection methods for nutritional attributes, methods including NIRS, HSI, RS, NMR, X-ray, and those based on acoustic properties, mechanical properties, and E-tongues are mainly used (Table 3).

**Table 3.** Non-destructive detection methods for nutritional attributes of cereal grains.

| Detection Methods | Objects | Devices | References |
|---|---|---|---|
| NIRS | Proteins, starches, and amino acids | Unity SpectraStar 2500XL-spectrometer | [114,115] |
| HSI | Proteins, oleic acids, and starches | OCI-UAV-1000 hyper-spectrometer | [31,116,117] |
| RS | Proteins, starches, amino acids, and oils | Renishaw Raman spectrometer | [118–122] |
| NMR | Oils | Minispec mq20 NMR spectrometer | [123] |
| X-ray | Trace elements | Hard X-ray microprobe | [71,124,125] |
| Acoustic properties | Proteins and ashes | Physical property analyzer | [36] |
| Mechanical properties | Proteins and starches | CT3 physical property analyzer | [50,63] |
| E-tongues | Starches and proteins | Self-made three-electrode E-tongue | [126] |

Grains show different spectral response characteristics in the near-infrared region when containing different contents of nutritional ingredients, so the nutritional attributes can be quantified based on spectral features. Shi et al. [114] collected NIR spectra of 48 wheat samples at 680 to 2500 nm and established the optimal NIR model between 1400 and 2500 nm using mean-centered partial least square regression (PLSR). The $R^2$ of crude proteins is predicted to be 0.98. Amanah et al. [115] determined diffuse reflectance spectra of 310 soybean samples in the waveband ranging from 1000 to 2500 nm and studied characteristic absorption bands of different nutritional ingredients of soybeans. In addition, they also built the PLS multi-variate analysis model for isoflavones and oligosaccharides in soybeans, and the predicted $R_2$ scores of the model for isoflavones and oligosaccharides are separately 0.80 and 0.72. Nutritional ingredients also differ in different varieties of grains in a same type, so the different varieties can be discriminated according to differences in nutritional attributes. Through principal component analysis (PCA), Wadood et al. [127] screened the spectral variables at 950 to 1650 nm and established a classification model for linear discriminant analysis (LDA). The model shows the highest classification accuracies of 100% and 73% separately for the origin and production year of wheat, and the highest classification accuracy of 98.2% for the genotypes of wheat grains. Fan et al. [128] coupled machine learning methodologies with a spectral dimensionality reduction algorithm to establish an assessment model for seed vigor, the accuracy of

which exceeds 84.0%. Based on PCA, Basati et al. [129] classified healthy and unhealthy wheat kernels at the accuracy of 100%, and the two types of wheat kernels are more effectively discriminated at wavelengths of 839 nm, 918 nm, and 995 nm. Zhang et al. [130] eliminated interference wavelengths successively using the equidistant combination-partial least squares and wavelength step-by-step phase-out PLS. On this basis, the constructed novel purity analysis model of rice seeds shows a predicted correlation coefficient of 0.920. Schütz et al. [131] established a global classification model for the geographical origin of grain maize by adopting the SVM classification technique, which achieved an overall classification accuracy of 95%. Qiu et al. [132] differentiated viable and unviable maize seeds based on spectral data within the waveband ranging from 1000 to 2500 nm and the established partial least square-discriminant analysis (PLS-DA) classification model shows the classification accuracy of 98.0%. By using the random forest algorithm, Wang et al. [133] screened spectra of soybeans in the range of 5793 to 12,489 cm$^{-1}$ and selected important variables to establish the classification model, which achieved an 80% classification accuracy of soybeans. Kusumaningrum et al. [134] used the variable importance in projection method to screen 146 optimal wavelengths from 1000 to 2500 nm and built the soybean seed viability prediction model combining PLS-DA. The model shows the prediction accuracy approaching 100% for viable and unviable soybean seeds.

HSI acquires spectral information when collecting image information, so it can be used to detect the nutritional attributes of grains. HSI allows segmentation of hyperspectral data from single grain kernels and also can segment hyperspectral data of different parts of single grain kernels, so it exhibits unique advantages in quality analysis of single grain kernels. Caporaso et al. [31] collected hyperspectral images of individual wheat kernels from 980 to 2500 nm and the $R^2$ scores for the calibration and validation sets of the established detection model for the protein content based on PLSR analysis are separately 0.82 and 0.79. Fu et al. [116] selected 23 characteristic wavelengths of oleic acids and 21 characteristic wavelengths of linoleic acids in hyperspectral images of soybean kernels in the range from 600 to 1000 nm. On this basis, they established the PLS-DA classification model of different oil contents, which shows the classification accuracies separately of 98.6% and 86.7% for the calibration and validation of oleic acids. Liu et al. [117] sampled hyperspectral images above and below the embryos of individual maize kernels in the range from 930 to 2500 nm and used the competitive adaptive reweighted sampling (CARS) method to select characteristic wavelengths. They established the PLSR and non-linear statistical data model using the ANN algorithm and its predicted $R$-value for maize starch content is 0.96. Weng et al. [135] collected hyperspectral images from 10 high-quality paddy varieties in China at 400 to 1000 nm. The classification model for paddy varieties based on PCA networks using spectral and morphological features exhibits classification accuracies separately of 98.66% and 98.57% in the training and prediction sets. Guo et al. [136] obtained hyperspectral images of the mixture of rice produced in Wuchang City and Jiangxi Province, China from 380 to 1000 nm and the established full-wavelength SG1-PLSR model shows a predicted $R$-value of 0.9909. The $R$-value of the simplified PLSR model established based on 15 important wavelengths selected according to the weighted regression coefficient is 0.9769. Miao et al. [137] imaged 800 maize kernels in eight varieties in the range from 386.7 to 1016.7 nm, and the $t$-distributed stochastic neighbor embedding ($t$-SNE) model pre-processed via PCA exhibits the highest classification accuracy of 97.5%, which is far higher than other models (75% for PCA, 85% for kernel principal component analysis, and 76.25% for local linear embedding). Li et al. [54] collected 3600 hyperspectral images of soybeans in the range from 866.4 to 1701.0 nm and established the one-dimensional CNN model of soybeans. The classification accuracies of the model for four soybean varieties (Zhonghuang 13, 37, 39, and 57) exceeded 98%.

The molecular vibration and rotation information can be obtained based on Raman spectra of nutritional ingredients in grains at different frequencies of incident light, thus realizing the quantitative detection and qualitative analysis of the quality [138,139]. RS can be used for qualitative analysis such as the varietal classification of grains including wheat,

paddy, and maize. Liu et al. [118] collected Raman spectra of eight wheat varieties with waxy proteins from 700 to 2000 cm$^{-1}$, measured spectra in several specific regions, and established the discriminant analysis (DA) classification model. The accuracies of the model in the calibration and validation sets are 94.4% and 94.6%, respectively. Zhu et al. [119] acquired Raman spectra of 107 paddy varieties in China from 200 to 1500 cm$^{-1}$, used PCA for preliminary evaluation, and established the soft independent modelling of class analogy model. The model can identify high-quality and poor-quality rice at an accuracy of between 80 and 100%. Liu et al. [140] collected Raman spectral data from 760 maize seeds from four different varieties from 400 to 1800 cm$^{-1}$, selected characteristic wavelengths through CARS and established the MCARS-genetic algorithm-SVM model in 13 wavebands. The calibrated classification accuracy and prediction accuracy of the model are 99.29% and 100%, respectively. The grain quality can be quantitatively assessed based on characteristic peaks on Raman spectra of different nutritional ingredients in different grains. Pezzotti et al. [120] proposed and verified the quantitative algorithm for calculating the amylose and amylopectin components based on spectral differences across 830 to 895 cm$^{-1}$. They also constructed the Raman equation for quantitatively determining the phenylalanine (PHE) and tryptophan components based on the significant features of PHE at 1003 cm$^{-1}$ and the breathing vibration of indole rings of tryptophan at 768 cm$^{-1}$. Singh et al. [121] collected transmitted Raman spectra of soybean flour from 400 to 1950 cm$^{-1}$ and established the multi-variate linear regression classification model based on the spectral measure. The predicted $R_2$ scores of proteins and fats in the soybean flour are separately 0.87 and 0.87. Yang et al. [122] studied Raman spectral characteristics of maize seeds from 380 to 180 cm$^{-1}$ according to samples of standard chemical components of maize, including maize starch, protein, oil, and bran. They found that the characteristic peaks at 477, 1443, 1522, 1596, and 1654 cm$^{-1}$ in their Raman spectra are separately correlated to the maize starch, oil-starch mixture, zeaxanthin, lignin, and oil.

Based on the NMR principle, a series of detectable inductive signals with attenuated intensity are generated when exciting atomic nuclei with fixed magnetic moments in samples using a specific impulse train. The intensity of the relaxation signals is in direct proportion to the number of nuclear spins in the detected samples (a quantitative basis) and the signal attenuation process is closely related to the constituent structure of the detected substances (a qualitative basis). The inverse analysis of signals using a mathematical method can yield information about various components and microstructures that fail to be obtained using other approaches, thus realizing the detection objective. Li et al. [123] identified the mass and oil content of individual maize kernels using an NMR sorting system. The mass of individual maize kernels is listed (in descending order) as haploid, diploid, and aborted kernels on the whole, with an extremely significant difference between diploid and aborted kernels and insignificant differences among other kernels; the oil contents are listed (in descending order) as diploid, haploid, and aborted kernels, and the oil contents are distributed in gradients in the three types of kernels.

In X-ray fluorescence spectrometry, various elements in samples release X-ray fluorescence with characteristic energy after being excited by irradiating these samples using X-rays. By measuring the wavelength and intensity of these X-rays, the elements present in the samples can be quantified. X-rays can be utilized to detect trace elements in grains. Lemmens et al. [124] adopted pearling, synchrotron X-ray fluorescence microscopy mapping, and X-ray absorption near-edge structure imaging of wheat to reveal changes in mineral elements such as Zn, Fe, and S during germination of wheat. This provides an opportunity for improving the nutritional quality in the food processing process. Chen et al. [125] investigated wheat flour samples by using the energy dispersive X-ray fluorescence spectrometry and extracted characteristic variables using the CARS. They also identified 12 energy variables (corresponding to mineral elements) to characterize the geographical attributes of samples. Through PCA and quadratic discriminant analysis, they established a non-linear model, which has the identification accuracy of 84.25% for origins of wheat flour through five-fold cross validation. Nguli et al. [71] estimated concentrations of Mn, Fe, Cu, and

Zn in soybean kernels based on total reflection X-ray fluorescence, and their results show that soybean contains sufficient trace elements. Except for Fe which is present in high concentrations, concentrations of other elements are relatively low yet within the necessary range.

Detection based on acoustic features is used to evaluate grain quality according to the vibration responses of grains under certain excitation. The interactions between acoustic vibration and grains include reflection, scattering, transmission, and absorption. The propagation mode of acoustic vibration is determined by the vibro-acoustic characteristics of grains, which are related to the mechanical and structural properties of grains. Responses of grains to vibration are dependent on the elastic modulus, Poisson's ratio, density, mass, and shape of grains, and the mechanical and structural properties of grains change with the growth and metabolism of cells. Detection based on acoustic features is widely used to assess attributes that are related to the mechanical and structural properties [93]. Khorshidi et al. [141] detected wheat varieties through transmission of longitudinal ultrasonic waves at five frequencies and established the PCA and generalized linear model thereof. Their results imply that, compared with other frequencies, ultrasonic measurement at 10 MHz can better identify wheat varieties and the ultrasonic phase velocity and longitudinal storage modulus at 10 MHz are the optimal discrimination factors for wheat varieties. Khorshidi et al. [141] also verified the capacity of low-intensity ultrasound at 10 MHz in identifying wheat varieties using 23 red, hard, spring wheat varieties. They confirmed through PCA that the ultrasonic parameters are significantly correlated to the protein content, falling number, and grain ash content of wheat. An acoustic detection method can be applied to quality detection and variety classification of grains, while it is susceptible to external environmental interference and can only be applied to a limited number of scenarios.

Mechanical properties of grains refer to mechanical features thereof under all types of applied load. Determining the mechanical parameters (rupture force, rupture energy, etc.) of grains under load can reduce the loss of grains in the harvest, transportation, storage, and processing processes. Research into the organizational structures and mechanical properties of grains provides a certain physical basis for improving the nutritive value and taste of grain products. Mechanical properties of grains are also an important factor that determines their processing quality, and determining the type and calculating the value of load borne by grains in the processing process can provide a basis for precision-design of relevant agricultural machinery [95]. Feng et al. [63] designed an orthogonal rotation combination test scheme. By using a multi-parameter controllable thin layer drying test bench, they dried maize under different conditions (temperatures of 30 to 60 °C, relative humidity of 30% to 60%, air velocities of 0.46 to 0.94 m s$^{-1}$, initial moisture contents of maize of 20% to 30% w.b., and tempering ratios of 0 to 3). Thereafter, they employed a texture analyzer to measure various mechanical properties of dried maize kernels, including the rupture force, rupture energy, elastic modulus, and brittleness. They then established the model for relationships of the rupture force, rupture energy, elastic modulus, and brittleness with the drying conditions. As the drying temperature is increased from 30 °C to 60 °C, the rupture energy, elastic modulus, and brittleness of maize kernels separately are increased by 19.11%, 11.76%, and 4.02%; when the relative humidity during drying is increased from 30% to 60%, the rupture force, rupture energy, elastic modulus, and brittleness separately are increased by 15.07%, 13.74%, 20.73%, and 3.31%. Delwiche et al. [50] explored the relationships of super-soft wheat with three compressive strength properties (maximum stress, Young's modulus, and Poisson's ratio), with the correlation coefficients separately being 0.76, 0.66, and 0.75. Grain quality based on mechanical properties was detected mainly on the basis that the quality distribution of grains influences organizational structures of grains, which are then shown as the difference in mechanical properties. At present, the correlation between nutritional ingredients and organizational structures of grains remains unclear, and the grain quality detection based on mechanical properties is still in the initial exploratory stage.

Lu et al. [126] measured the original voltametric signals of rice-flour slurry using a self-made E-tongue and built a CNN+BpNN model. The training and prediction accuracies of the model for physical and chemical indicators including the degree of chalkiness, amylose, protein, starch, and total metal content are separately between 84.3% and 92.0% and 81.9% and 89.5%. Zhao et al. [40] assayed rice filtrate by using an E-tongue ($\alpha$-Astree, and Alpha MOS, France). Discrimination factors DF1 and DF2 separately account for 97.24% and 97.58% of the total variability of paddy in the storage period, and the DF1 value of storage devices shows an increase trend while the DF2 value first increases, then decreases. The results show that rice samples under different storage conditions can be distinguished. Guo et al. [142] acquired current response signals of wheat leachates during five different storage periods using a self-made large amplitude pulse voltammetry (MLAPV) based E-tongue system. They also built a detection model based on wavelet packet transform, improved artificial fish swarm algorithm, and extreme learning machine. The model achieved the training and testing accuracies separately of 96% and 92% for aged wheat stored for five different periods.

NIRS, HSI, and RS can all be used to detect multiple quality indices; NMR detection can detect oils; X-rays can be employed to detect trace elements. In comparison, acoustic and mechanical responses of grains are insignificantly correlated with the nutritional attributes of grains; because grain leachates need to be prepared when detecting grains using an E-tongue, application of E-tongues is, to some extent, limited.

*2.3. Non-Destructive Inspection Methods for Safety Attributes of Cereal Grains*

Regarding safety attributes, the maximum limits for pesticide residues, fungal toxins, and contaminants in grains have been listed in the Chinese National Standards for Food Safety. Traditionally, gas chromatography and liquid chromatography are commonly used for determining the maximum limits of pesticide residues and fungal toxins, and atomic absorption spectrometry (ABS) is utilized to measure heavy metal residues. Modern methods for detecting safety attributes mainly include NIRS, HSI, RS, X-ray, and E-noses (Table 4).

**Table 4.** Non-destructive inspection methods for safety attributes of cereal grains.

| Detection Methods | Objects | Devices | References |
|---|---|---|---|
| NIRS | Fungal toxins | Zeiss fiber optical spectrometer | [143,144] |
| HSI | Fungal toxins | ANDOR EMCCD camera + Xenics LWNIR camera | [145] |
| RS | Pesticide residues and fungal toxins | 1064-nm NanoRam Raman spectrometer | [32,60,146] |
| X-ray | Heavy metals and fungal toxins | Y. CHEETAN micron-resolution X-ray CT scanner | [111,147] |
| E-noses | Pesticide residues and fungal toxins | Fox3000 E-nose | [148,149] |

NIRS can be used to detect fungal toxins. Jiang et al. [143] collected spectra of wheat from 600 to 1600 nm and adopted the LDA and PLSR to construct a qualitative and quantitative analysis model for the degree of infection of wheat infected with harmful mold. The LDA model shows an overall identification rate above 90.0% for wheat samples with different degrees of mildew, and the determination coefficient of the PLSR model reaches 0.8592. Ong et al. [144] proposed a modified simulated annealing (MSA) algorithm to determine the concentration of aflatoxin B1 in rice. Compared with the full-spectrum regression model, the MSA algorithm provides more accurate detection (an improvement of more than 44% at the low concentration and one exceeding 62% at the high concentration).

HSI that combines spectral and image information can reach a higher detection accuracy. Zhou et al. [145] collected hyperspectral images of maize flour samples with different aflatoxin B concentrations from 430 to 2400 nm and proposed two algorithms with which to calculate the inter-class variance ratio and weighted inter-class variance ratio to extract the characteristic wavelengths. In this way, they established the SVM model, the average classification accuracy of which reaches 96.18%.

RS has been extensively used to detect safety attributes of grains [150–153]. Pesticide residues in grains produce clear characteristic peaks in the Raman spectra and changes in grain quality caused by mildew may induce variation of Raman spectra. According to response characteristics of Raman spectra of pesticides and mildew, the safety attributes of grains can be quantitatively detected. Qiu et al. [60] obtained Raman spectra of healthy wheat kernels, and wheat kernels with slight and severe gibberellic disease from 240 to 1736 cm$^{-1}$. A classification model was established by using an inception-attention network, with classification accuracies of 97.13%, 91.49%, and 93.62% separately in the training, validation, and prediction sets. Long et al. [146] acquired Raman hyperspectral images from 732 to 1007 nm, from which 50 variables were screened by combining CARS with variance features. These variables are more applicable to determine the fungal spore count of individual maize kernels. In addition, they established an optimal detection model based on PLSR, with an *R*-value of 0.8619 in the validation test. Weng et al. [32] prepared efficient gold nanorods for detection based on surface enhanced RS and resolved Raman fitting using the density functional theory. When the residual concentrations of the wheat leachate and contaminated wheat are separately 0.2 and 0.25 mg/L, the Raman signals of pirimiphos-methyl can be detected. Jiang et al. [154] collected the surface enhanced Raman spectra of paddy samples with different Dursban concentrations. The *R*-values of calibration and prediction sets predicted using the CARS-PLS model are separately 0.9942 and 0.9881, and the LOD is 0.01 mg/mL.

X-rays are fluorescent and have been widely used to explore heavy metal residues in grains [92]. Sang et al. [147] developed an energy dispersive X-ray fluorescence spectrometric method to detect cadmium in rice, which resulted in an *R*-value of 0.9814 when compared with the measurement results of graphite furnace atomic absorption spectrometry. Zhou et al. [111] scanned wheat kernels through micron-resolution X-ray CT and determined whether wheat kernels are infected with mold or not by comparing grey values in different states.

E-noses, also known as artificial olfactory systems, are electronic systems that imitate biological noses. They mainly identify different types and components of materials according to odor. Mildew and pesticide residues of grains may produce volatile substances in the storage process, which can be qualitatively studied and quantitatively detected using an E-nose [97,155,156]. Zhou et al. [64] cultivated insect pests in a container containing 1 kg of rice at 15 °C and 30 °C for four weeks and detected the volatile substances in the rice with insect pests using an E-nose (Cyranose 320) every five days. In addition, the PCA was adopted to build the classification model. Results show that the E-nose can accurately identify clean rice and rice with insect pests at 30 °C, while it fails to discriminate among those stored for one, two, and three weeks. Mishra et al. [67] used an E-nose to evaluate changes in the quality of wheat artificially infected with *Rhyzopertha dominica* to different degrees in four storage periods. They also optimized the sensor array via hybrid neuro-fuzzy-assisted electronic nose analysis and detected the number of insect pests, uric acid content, and protein content, with *R* values of 0.999, 0.985, and 0.973 respectively. Ku et al. [148] inoculated *Aspergillus candidus* in fresh paddy, which was then loaded in a simulation grain bin and detected using an online detection system for mildew of grains. A model for the mold content was established by optimizing parameters using the particle swarm optimization. The $R^2$ score for mold detection and the minimum LOD for mold content are 0.9839 and $1.5 \times 10$ cfu/g, respectively. Zhao et al. [149] collected the odor responses using an E-nose from wheat inoculated with mold after storage for 0, 1, 3, 5, and 7 days, and they established the relationship model with the degree of infection of mold. The PCA model can accurately differentiate wheat samples without mildew and with slight and severe mildew. The identification rate of the LDA model for the mildew degree of wheat infected with a single mold exceeds 90.0% and that for all wheat samples reaches 84.0%. The predicted $R^2$ of the PLSR model for the total fungal count is 0.852.

NIRS and HSI can qualitatively identify and quantitatively detect fungal toxins. RS can detect fungal toxins and pesticide residues according to different Raman characteristic

peaks. The mildew and heavy metal residues in grains can be detected based on X-rays due to their depth of penetration and fluorescent characteristics. E-noses can detect safety attributes of grains according to the volatile gases produced by fungal toxins and pesticide residues.

## 3. Unsolved Technical Problems

For different quality indices of grains, different types of detection methods can be used to realize high-accuracy detection. However, these methods still have room to improve in terms of grain-quality detection.

1.  High cost. ① Optical detection methods including NIRS, HSI, and RS have developed to relative maturity, while the detection cost of full-spectrum devices is high. ② NMR instruments also face the problem of high cost.
2.  Environmental interference. Grain quality detection based on acoustic and thermal features is greatly influenced by the environment (ambient noise and temperature).
3.  Detection principle. ① The relationship between the mechanical features and quality indices of grains remains unclear. ② Water is an important factor that affects the dielectric property of grains, while the relationship between quality indices and dielectric properties of grains is also poorly understood. ③ Detection objects of E-tongues must be grain leachates, which limits the application thereof to the quality detection. ④ X-rays may contaminate grains.
4.  LOD. E-nose detection is limited by the LOD of gas sensors and the method fails to identify problems including early mildew in grains.
5.  Moisture detection. Water is an important factor that influences grain quality and must be detected in all stages including the harvest, storage, trading, transportation, and processing of grains.
6.  Grading and classification of grains. Grains can be graded and classified according to differences in multiple quality indices of grains in accordance with the national or international standards. However, research on grain quality using a single detection technique can only determine one or several indices, which fails to meet the grading and classification demands imposed in practice.
7.  Practical application issues. In different applications, grain quality inspection equipment faces different challenges; for example, the aerodynamic characteristics of the grain seeds during sowing, and the vibration of the machinery during harvesting can affect the quality inspection results.

## 4. Future Research Directions

1.  ① Considering that screening of characteristic wavelengths of different quality indices of grains is still an important part in existing quality detection research, device development based on characteristic wavelengths can substantially reduce the cost of analysis. This is conducive to the popularization and application of optical detection devices. In recent years, NIRS spectrometers [157,158] have also been upgraded with the development of NIRS analysis and chemometrics methods [159–162]. Liu et al. [163] selected four characteristic wavelengths to develop the portable near-infrared quality detector, which can realize the real-time determination of proteins and moisture in wheat kernels. The development of multispectral imaging [164] based on characteristic wavelengths can overcome this problem. Sendin et al. [165] discriminated between high-quality and poor-quality maize using 19 characteristic wavelengths, with the classification accuracy in the range of 83% to 100%. ② NMR can be divided into high-field and low-field ones [166]. A high-field NMR spectrometer contains expensive superconducting magnets, so it has complex structures and its signals are difficult to process. A low-field NMR spectrometer uses low-cost permanent magnets, so it is preferred in grain-quality detection.
2.  Acoustic features include audible sound and ultrasonic waves. Detection based on audible acoustic features is susceptible to the ambient noise, while acoustic detection

based on ultrasonic waves can avoid environmental interference, and is an important method of applying acoustic methods to grain-quality detection [36]. Eliminating interference (including the effect of changes in ambient temperature) with detection based on thermal features is the top priority for improving the accuracy of detection of grain quality. Mangus et al. [62] performed environmental calibration using a temperature reference plate, which compensates for environmental influences including air temperature, relative humidity, solar radiation, and camera temperature, thus maintaining the measurement temperature.

3.　① The difference in mechanical properties of cereal grains is determined by the tightness of bonding of main components including starches and proteins therein. By using an electronic universal testing machine, Cheng et al. [167] measured the shear resistance of wheat. In this way, they obtained that the shear resistance is significantly positively correlated with the protein content, positively correlated with wet and dry gluten, negatively (albeit insignificantly) correlated with the starch content, and positively (albeit insignificantly) correlated with the bulk weight and thousand-kernel weight. Existing research into the mechanical properties of cereal grains mainly focuses on grain quality evaluation [168], while the correlation of these properties with quality indices remains to be further studied. ② Moisture detection of grains based on dielectric properties has reached an extremely high accuracy, so developing detection devices applicable to different application scenarios is a potential direction for future development [169]. ③ Preparation of grain leachates is laborious and the detection electrodes of E-tongues need to be cleaned and polished in a complex process before the next detection [142]. Therefore, the development of preparation techniques of detection samples for E-tongues and the upgrading of detection electrode materials are the only way to realizing real-time efficient detection of quality indices of grains. ④ X-rays include hard and soft variants [170]; hard X-rays may damage grains, while soft ones have low penetrability and therefore, they are applicable to detection of quality indices of grains.

4.　The multi-scale, systematic organization for imitating biological noses through bionics design is one of the methods to improve the LOD of E-noses. In addition, development of high-LOD gas-sensing materials is also an important approach to improving the performance of E-noses [171].

5.　Research into moisture detection in grains has developed to relative maturity, and high-accuracy moisture detection can be realized based on dielectric properties and NMR [56,75,172–175]. Robust progress has been made in moisture detection devices based on dielectric properties [79,169,176], while those based on NMR have not developed to any substantial extent. Moisture detection devices for grains in different application scenarios should be a focus of future research.

6.　The combination of multiple quality detection techniques of grains can detect multiple quality indices in the grading and classing requirements, thus realizing grain grading and classification. At present, some studies have combined multiple detection techniques to improve the detection accuracy [177–181]. Realizing the grading and classification of cereal grains by combining multiple quality detection techniques remains the focus of future research.

7.　To guarantee the normal operation of grain quality detection equipment, the interference factors affecting grain quality detection equipment are studied for specific application scenarios. Gierz et al. [182] collected seed image data at the air velocity (15, 20, 25 m/s) of a pneumatic seeder pipeline conveying seeds based on the aerodynamic characteristics of grain seeds [183,184], and constructed a classification model based on multilayer-based perceptron network, which has a correct classification coefficient of 0.99 for contaminants in seeds at a sowing speed of 15 m/s. Studying the influencing factors in different application scenarios is necessary to promote the application of grain quality detection instruments.

**Author Contributions:** Conceptualization, Y.L. and J.N.; investigation, Y.L., J.Z., H.Y. and M.S.; writing—original draft preparation, Y.L.; writing—review and editing, Y.L., J.Z. and J.N.; visualization, Y.L., J.Z., H.Y. and M.S.; supervision, Y.Z., W.C., X.J. and J.N.; funding acquisition, Y.Z., W.C., X.J. and J.N. All authors have read and agreed to the published version of the manuscript.

**Funding:** This research was funded by the National Key Research and Development Program of China (grant number 2021YFD2000101), Fundamental Research Funds for Central Non-profit Scientific Institution (grant number S202219), National Natural Science Foundation of China (grant number 31871524), Modern Agricultural Machinery Equipment & Technology Demonstration and Promotion of Jiangsu Province (grant number NJ2021-58), and the Primary Research & Development Plan of Jiangsu Province of China (grant number BE2019306, BE2021304).

**Data Availability Statement:** Data sharing not applicable. No new data were created or analyzed in this study. Data sharing is not applicable to this article.

**Acknowledgments:** We would like to thank all the researchers in the Intelligent Equipment Research Group of the National Engineering and Technology Center for Information Agriculture and all the foundations for this research.

**Conflicts of Interest:** The authors declare no conflict of interest.

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
