# Peer review of "Non-Destructive Quality-Detection Techniques for Cereal Grains: A Systematic Review"

_agronomy, doi:10.3390/agronomy12123187_

Round 1
Reviewer 1 Report (Previous Reviewer 3)
I am happy with the revised manuscript and would recommend it for publication.
Author Response
Thank you for your comments on our work. We have checked the English language and style again.
Reviewer 2 Report (New Reviewer)
The article looks good to me. However, the following suggestion would contribute to the article's quality.
1. Add the organization section of the article.
2. Add the implications section of the article.
3. If possible, please raise the quality of the figures. For example, in figure 2, some values/words are unclear.
4. Don't separate the tables into 2 pages, unless impossible. For example, Table 3.
5. Make sure that all content inside the table is the same are aligned center.
6. I think the usage of symbols in some sections is inappropriate.
7. Where is the methodology section for the systematic review?
Author Response
Q1 & Q2: We have added the organization section and implications section in lines 119-137 of the paper.
Q3: We replace the figures with bmp format to ensure the values/words are clear enough.
Q4: We have adjusted all the tables to ensure that they are not separated into 2 pages.
Q5: We have checked and adjusted the contents of the table, and make sure that all content inside the table is the same are aligned center.
Q6: Thanks for your suggestion. we checked the usage of symbols and modified some inappropriate ones.
Q7: Thanks for your advice. we have added the methodology section in lines 60-84 of the paper.
Reviewer 3 Report (New Reviewer)
The paper provides a comprehensive review of the application of existing methods for the analysis of grain properties from different aspects. The references reviewed consider SOTA including most recent work. Please improve the following:
1- The introduction should include the methodology of the review? How were then SOTA found? What search engines were used? Exactly what terms were searched for? How are the SOTA examined? etc. Otherwise, it would not be a "systematic review", literally.
2- Too little is done in the introduction to give a general introduction to the subject, and instead too much detail is given too early about the existing related work.
3- Sections do not present their content in a structured manner. Using subsections and clustering related content definitely improves the paper.
4- The paragraphs are also not well organized. Most of them are very long and contain multiple ideas collected in one place. Please review the division of your paragraphs and limit each paragraph to only one idea. Next idea; next paragraph.
5- Conclusion is missing! I would really like to read the conclusions of the authors who have gone through many SOTA on this subject and see from their point of view how they come to their conclusions?
Good luck
Author Response
Q1: Thanks for your advice. we have added the methodology section in lines 60-84 of the paper. We used the Web of Science search engine to find SOTA in the core collection database. We searched the references by combining the keywords cereal, grain, and quality with the technical words of physical properties such as near-infrared spectroscopy, hyperspectral imaging, Raman spectroscopy, optical, dielectric, nuclear magnetic resonance, X-ray, electromagnetic, acoustic, thermal, and mechanical, and sensory features such as electronic eye, computer vision, electronic nose, electronic tongue, and sensory, respectively. We classified all the references according to the quality as the first condition and the techniques as the second condition, and summarized and analyzed the research purposes, contents, methods and results of each reference.
Q2: We have added the general introduction in lines 60-84 and 119-137 and removed the description of the existing related work.
Q3 & Q4: We have added a block diagram of the article organization (Figure 3) and briefly described the four parts of the content of the article in lines 123-137. and we have added subsections and have redefined the paragraphs of the article.
Q5: Thanks for your suggestion. In the third part of the paper, we summarized the conclusions of the references and combined with our analytical screening to obtain the unsolved technical problems of SOTA. In the fourth part, we followed the perspective of the authors' conclusions and combined our own analysis to get the future research directions.
Round 2
Reviewer 2 Report (New Reviewer)
The authors revised the comments accordingly.
Please proceed with accepting the article in its present form.
This manuscript is a resubmission of an earlier submission. The following is a list of the peer review reports and author responses from that submission.
Round 1
Reviewer 1 Report
The work presented to me for review is a very valuable study on non-destructive methods of grain evaluation. All methods were characterized very well and the directions for their potential development were set. I have no objections to the work and recommend it for publication in its current form.
Reviewer 2 Report
The work contains a lot of valuable information, but no attention was paid to the aerodynamic properties of the kernels. Pneumatic seeders are used more and more often, where these properties are of great importance.
When analyzing the literature, you can find the following publications:
1. "Assessment of physical and aerodynamic properties of corn kernel (KSC 704)"
2. "The method and a stand for measuring aerodynamic forces in every plane on the basis of an image analysis" and many others "
The work of prof. Boniecki pt. "Detection of the granary weevil based on x-ray images of damaged wheat kernels."
Perhaps it is also worth paying attention to the work: The Use of Image "Analysis to Detect Seed Contamination — A
Case Study of Triticale "," Validation of an image-analysis-based method of measurement of the overall dimensions of seeds "and" The application of optoelectronic elements to control the sowing process "
I hope my tips will enrich this manuscript.
Reviewer 3 Report
Yiming Liu et al article includes a hot topic, however the paper requires major revisions.
1) The structure of the paper needs improvement as it lacks clarity.
2) The limitation of each study included in the review should be highlighted in the listed tables or if required a table can be separately added for better understanding
3) Spelling and grammatical mistakes need to be corrected.
4) Future research directions needs to be improved